# Learning Along the Arrow of Time: Hyperbolic Geometry for Backward-Compatible Representation Learning

**Ngoc Bui**[† 1]  **Menglin Yang**[1]  **Runjin Chen**[† 2]  **Leonardo Neves**[3]  **Mingxuan Ju**[3]  **Rex Ying**[1]  **Neil Shah**[3]  **Tong Zhao**[3]

## Abstract

Backward-compatible representation learning enables updated models to integrate seamlessly with existing ones, avoiding to reprocess stored data. Despite recent advances, existing compatibility approaches in Euclidean space neglect the uncertainty in the old embedding model and force the new model to reconstruct outdated representations regardless of their quality, thereby hindering the learning process of the new model. In this paper, we propose to switch perspectives to hyperbolic geometry, where we treat time as a natural axis for capturing a model's confidence and evolution. By lifting embeddings into hyperbolic space and constraining updated embeddings to lie within the entailment cone of the old ones, we maintain generational consistency across models while accounting for uncertainties in the representations. To further enhance compatibility, we introduce a robust contrastive alignment loss that dynamically adjusts alignment weights based on the uncertainty of the old embeddings. Experiments validate the superiority of the proposed method in achieving compatibility, paving the way for more resilient and adaptable machine learning systems.

## 1. Introduction

Representation learning has become an integral part of modern machine learning, offering a way to transform raw data—be it text, images, or other modalities—into compact representations or "embeddings" that encode their essential structure (Chen et al., 2022; Bengio et al., 2013; Guo et al., 2019). A high-quality embedding model can unlock

† Work done during an internship at Snap Inc. [1]Yale University, New Haven, CT, USA [2]The University of Texas at Austin, TX, USA [3]Snap Inc., Bellevue, WA, USA. Correspondence to: Rex Ying <rex.ying@yale.edu>, Tong Zhao <tong@snap.com>.

*Proceedings of the 42$^{nd}$ International Conference on Machine Learning*, Vancouver, Canada. PMLR 267, 2025. Copyright 2025 by the author(s).

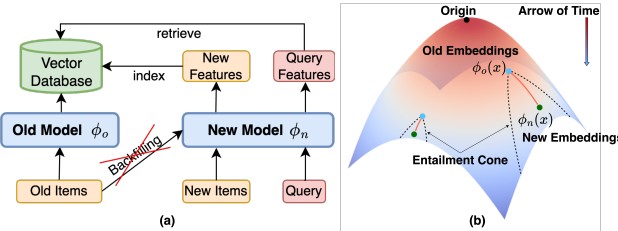

*Figure 1. a)* A typical setting of the backward-compatible training problem. A large gallery set is embedded and indexed into a vector database using the old model. Updating to the model may require re-indexing the entire vector database (*backfilling*). *Backward-compatible training* allows the new model to query and retrieve from the vector database of old embeddings directly. *b)* Simulating the model evolution in hyperbolic space with the entailment cone.

numerous applications: from semantic search (Zhu et al., 2023; Chandrasekaran & Mago, 2021) and recommendation systems (Liu et al., 2024; Ko et al., 2022) to retrieval-augmented generation in large language models (LLMs), where a vector database is used to find and integrate relevant information at inference time (Gao et al., 2023). The ubiquity of pretrained models such as BERT (Devlin, 2018), CLIP (Radford et al., 2021), or GPT-style architectures (Radford et al., 2018) makes these off-the-shelf embeddings readily available. Enterprises and researchers alike increasingly rely on these pretrained representations, adapting them to in-house data or tasks for streamlined development and fast deployment (Bommasani et al., 2021).

Yet, embedding models do not remain static. Model updates are often necessary to leverage more advanced architectures, integrate new domain knowledge, or address performance limitations (Wang et al., 2024). However, updating these models typically introduces significant challenges. In large-scale retrieval pipelines or scenarios where legacy embeddings have been indexed, these upgrades can be disruptive if older embeddings become incompatible with the new model's representations. Meanwhile, re-embedding all data not only incurs high computational and operational costs but also raises privacy concerns (Price & Cohen, 2019), especially in applications like face recognition (Chen et al.,

2019) and person re-identification (Cui et al., 2024). These issues highlight the need for *backward-compatible* representation learning: A model update is considered *backward-compatible* if embeddings from the new model can interact and retrieve information from embeddings generated by its predecessor (see Figure 1). This property allows old and new embeddings to coexist seamlessly without the need to reprocess sensitive or extensive datasets (Shen et al., 2020).

While several methods have been proposed for backward-compatible representation learning (Shen et al., 2020; Meng et al., 2021; Ramanujan et al., 2022; Zhou et al., 2023; Pan et al., 2023; Biondi et al., 2024; Jang & Lim, 2024), they concentrate on Euclidean or hyperspherical spaces powered by the cosine distance. A common strategy involves adding an extra loss term to align older and newer embeddings within the same space, compelling the new model to reconstruct the old representations regardless of their quality. Consequently, if the old embeddings are suboptimal due to the limited capacity of the old model, the new model's ability to learn richer or more accurate representations is constrained. This mismatch between strict compatibility and model evolution becomes especially relevant as models grow larger and more specialized, creating a larger gap between old and new representations.

We argue that hyperbolic geometry is a natural fit for describing model evolution due to its unique properties. *Firstly*, its exponential growth of areas and volumes with respect to radius makes it particularly suited for applications involving continual model updates as new entities or classes emerge. *Secondly*, recent studies (Khrulkov et al., 2020) show that hyperbolic encoders can effectively capture representation uncertainty, enabling 'free' uncertainty estimation along the time axis. This capability has been successfully applied to enhance performance across various tasks, including image segmentation (Atigh et al., 2022), action recognition (Franco et al., 2023), and video prediction (Surís et al., 2021). In the backward-compatible training problem, model updates can be seen as transformations of the embedding space over time, enabling new representations to evolve while maintaining consistency with their previous counterparts. This perspective aligns with the structure of Minkowski space-time (Stein, 1968), where hyperbolic geometry is used to describe the space-time curvature (Einstein et al., 2013; Barrett, 2011) and model dynamical systems where relationships among particles evolve over time (Araújo & Viana, 2008).

**Proposed work.** In this paper, we reframe backward-compatible representation learning through the unique lens of hyperbolic geometry and propose a novel framework, namely *Hyperbolic Backward-Compatible Training* (HBCT), to align new models with legacy representations. The core idea is to lift the embeddings into hyperbolic space

and use the time-like dimension as a natural axis to capture model evolution and the uncertainties inherent in vector embeddings. To capture the dynamic nature of embedding spaces, we employ the entailment loss (Ganea et al., 2018) to enforce a partial-order constraint between the old and new models. This constraint requires the new embeddings to remain within the entailment cone of their older counterparts, maintaining backward compatibility while embracing the improved expressiveness brought by new data or architectures. We further introduce a robust hyperbolic contrastive loss that dynamically adjusts alignment weights based on the uncertainty levels of old embeddings, enabling more adaptive and reliable alignment across model generations. To the best of our knowledge, this is the first attempt to leverage embedding geometry for maintaining consistency in updating models.

Through extensive experiments across diverse settings, we demonstrate that HBCT effectively enhances backward compatibility without compromising the performance of new embedding models. Specifically, HBCT improves CMC@1 compatibility of the new model by 21.4% compared to the strongest Euclidean baseline, and yields a 44.8% increase in mAP compatibility.

## 2. Preliminaries

### 2.1. Riemannian Manifolds and Hyperbolic Geometry

A *smooth* $n$-dimensional manifold $\mathcal{M}$ is a topological space where every point $\mathbf{x} \in \mathcal{M}$ has a neighborhood $U_{\mathbf{x}} \subseteq \mathcal{M}$ that is diffeomorphic (i.e., resembles) a neighborhood $V_{\mathbf{x}} \subseteq \mathbb{R}^n$. Each point $\mathbf{x}$ has an associated *tangent space* $\mathcal{T}_{\mathbf{x}}\mathcal{M}$, which is an $n$ dimensional vector space serving as a first-order local approximation of $\mathcal{M}$. A *Riemannian metric* $\mathfrak{g}$ on $\mathcal{M}$ is a smoothly varying collection $\mathfrak{g} = (\mathfrak{g}_{\mathbf{x}})_{\mathbf{x} \in \mathcal{M}}$ of positive definite bilinear forms $\mathfrak{g}_{\mathbf{x}}(\cdot, \cdot) : T_{\mathbf{x}}\mathcal{M} \times T_{\mathbf{x}}\mathcal{M} \to \mathbb{R}$. These bilinear forms determine the curvature at $\mathbf{x}$, which measures how $\mathcal{M}$ deviates from the flatness at that point. A *Riemannian manifold* is the pair $(\mathcal{M}, \mathfrak{g})$. Euclidean space $\mathbb{R}^n$ can be viewed as a Riemannian manifold with the standard Euclidean inner product and zero curvature. In contrast, *hyperbolic spaces* are Riemannian manifolds with *constant negative curvature*. This negative curvature leads to an exponential growth of areas and volumes with radius, different from the polynomial growth in Euclidean space.

### 2.2. Lorentz Models of Hyperbolic Geometry

There are several models that can represent $n$-dimensional hyperbolic spaces isometrically, such as the Poincaré ball model (Nickel & Kiela, 2017) or the Klein model (Mao et al., 2024). Following recent work (Desai et al., 2023; Bdeir et al., 2023), we use the *Lorentz model* (Nickel & Kiela, 2017) for its numerical stability (Mishne et al., 2023).

In what follows, we follow the notations and terminology from (Desai et al., 2023).

Lorentz model embeds $d$-dimensional hyperbolic space in $\mathbb{R}^{d+1}$ as a one-sheeted hyperboloid, where each point $\mathbf{x} \in \mathbb{R}^{d+1}$ can be written as $[x_{\text{time}}, \mathbf{x}_{\text{space}},]$, where $x_{\text{time}} \in \mathbb{R}$ acts like a "time coordinate" and $\mathbf{x}_{\text{space}} \in \mathbb{R}^d$ like "spatial coordinates". Let $\langle \cdot, \cdot \rangle$ be the usual Euclidean dot product. The *Lorentzian inner product* $\langle \cdot, \cdot \rangle_{\mathcal{L}}$ on $\mathbb{R}^{d+1}$ is given by

$$\langle \mathbf{x}, \mathbf{y} \rangle_{\mathcal{L}} = \langle \mathbf{x}_{\text{space}}, \mathbf{y}_{\text{space}} \rangle - x_{\text{time}} y_{\text{time}}. \quad (1)$$

An $n$-dimensional hyperbolic space of curvature $-K$, where $K > 0$, in the Lorentz model is defined as

$$\mathcal{L}^d = \left\{ \mathbf{x} \in \mathbb{R}^{d+1} : \langle \mathbf{x}, \mathbf{x} \rangle_{\mathcal{L}} = -\frac{1}{K}, \ x_{\text{time}} > 0 \right\}. \quad (2)$$

All points $\mathbf{x} \in \mathcal{L}^d$ satisfy $x_{\text{time}} = \sqrt{\frac{1}{K} + \|\mathbf{x}_{\text{space}}\|^2}$.

A *geodesic* between two points is the locally smooth and shortest path connecting two points. Geodesics in $\mathcal{L}^d$ are curves traced by the intersection of $\mathcal{L}^d$ with hyperplanes in $\mathbb{R}^{d+1}$ passing through the origin. The distance function between two points $\mathbf{x}, \mathbf{y} \in \mathcal{L}^d$ is

$$d_{\mathcal{L}}(\mathbf{x}, \mathbf{y}) = \sqrt{\frac{1}{K}} \ \cosh^{-1}\left( -K \langle \mathbf{x}, \mathbf{y} \rangle_{\mathcal{L}} \right). \quad (3)$$

The *tangent space* $\mathcal{T}_{\mathbf{p}} \mathcal{L}^d$ at a point $\mathbf{p} \in \mathcal{L}^d$ consists of all vectors in Euclidean space that are orthogonal to $\mathbf{p}$:

$$\mathcal{T}_{\mathbf{p}} \mathcal{L}^d = \{ \mathbf{z} \in \mathbb{R}^{d+1} : \langle \mathbf{p}, \mathbf{z} \rangle_{\mathcal{L}} = 0 \}. \quad (4)$$

To "move" between the manifold and its tangent space, we use the *exponential map* $\exp\mathbf{m}_{\mathbf{p}} : \mathcal{T}_{\mathbf{p}} \mathcal{L}^d \to \mathcal{L}^d$, defined as

$$\exp\mathbf{m}_{\mathbf{p}}(\mathbf{z}) = \cosh\left(\sqrt{K}\|\mathbf{z}\|_{\mathcal{L}}\right) \mathbf{p} + \frac{\sinh\left(\sqrt{K}\|\mathbf{z}\|_{\mathcal{L}}\right)}{\sqrt{K}\|\mathbf{z}\|_{\mathcal{L}}} \mathbf{z}. \quad (5)$$

Its inverse, the *logarithmic map* $\log\mathbf{m}_{\mathbf{p}} : \mathcal{L}^d \to \mathcal{T}_{\mathbf{p}} \mathcal{L}^d$, sends $\mathbf{x} \in \mathcal{L}^d$ back into the tangent space at $\mathbf{p}$:

$$\log\mathbf{m}_{\mathbf{p}}(\mathbf{x}) = \frac{\cosh^{-1}\left( -K \langle \mathbf{p}, \mathbf{x} \rangle_{\mathcal{L}} \right)}{\sqrt{\left( K \langle \mathbf{p}, \mathbf{x} \rangle_{\mathcal{L}} \right)^2 - 1}} \ \text{proj}_{\mathbf{p}}(\mathbf{x}), \quad (6)$$

where $\text{proj}_{\mathbf{p}}(\mathbf{x})$ maps a vector in ambient space $\mathbb{R}^{d+1}$ onto the tangent space at $\mathbf{p}$

$$\mathbf{z} + \text{proj}_{\mathbf{p}}(\mathbf{x}) = \mathbf{u} + K \mathbf{p} \langle \mathbf{p}, \mathbf{u} \rangle_{\mathcal{L}}. \quad (7)$$

These maps allow us to perform Euclidean operations in the tangent space while retaining the underlying hyperbolic structure once we map points back onto $\mathcal{L}^d$. Following the literature (Desai et al., 2023), we only use these maps at the origin of the hyperboloid $\bar{\mathbf{0}} = [\sqrt{1/K}, \mathbf{0}]^{\top}$.

## 3. Problem Statement

Following previous literature (Shen et al., 2020; Zhang et al., 2022a), we focus on a typical retrieval setting, where we have a large gallery $\mathcal{G}$ of items (*e.g.*, images) maintained in a vector database. Each item $x \in \mathcal{G}$ is associated with one class or identity $y \in \mathcal{Y}$, and is encoded into a dense vector by an embedding model $\phi(\cdot)$. During retrieval, each query $q \in \mathcal{Q}$ is encoded into a vector embedding, *i.e.*, $\phi(q)$, and relevant matches are retrieved by comparing with the embeddings in the gallery $\phi(\mathcal{G}) = \{\phi(x) \mid x \in \mathcal{G}\}$[1] using a distance function $d(\cdot, \cdot)$. The embedding model is trained on a dataset $\mathcal{D} = \{x, y\}$ with the base loss function:

$$\phi^{\star} = \arg\min_{\phi} \mathcal{L}_{\text{base}}(\phi, \mathcal{D}). \quad (8)$$

Here, we choose the loss function $\mathcal{L}_{\text{base}}$ to be the cross-entropy, following (Shen et al., 2020), but it could also be any other metric learning loss functions (Kaya & Bilge, 2019; Wang et al., 2018; Chen et al., 2020).

Let $\phi_o$ be the existing encoder trained on an old dataset $\mathcal{D}_{old}$ and $\Phi_o^{\mathcal{G}}$ is the existing gallery embeddings encoded using $\phi_o$. We now want to train a new encoder $\phi_n$ to replace $\phi_o$ subject to the arrival of new data $\mathcal{D}_{new} \supseteq \mathcal{D}_{old}$, more powerful architectures, or new pre-trained weights of existing architectures. Note that new data $\mathcal{D}_{new}$ may contain new identities or classes different from $\mathcal{D}_{old}$.

Our goal is to train the new model $\phi_n$ in a way that it could be directly compatible with the old model $\phi_o$: new queries encoded by $\phi_n$ should yield *similar or improved* retrieval performance on the existing gallery $\Phi_o^{\mathcal{G}}$ produced by $\phi_o(\cdot)$. This property enables a *backfill-free* approach—allowing the old embeddings to be gradually replaced by $\phi_n(x)$ as needed—while immediately benefiting from the new model's improved capabilities.

## 4. Methodology

As mentioned in the introduction, directly forcing new embeddings to reconstruct outdated representations from the old model can impair the learning process of the new model, because some old embeddings may be poorly represented. To address this, we propose to lift the embeddings to hyperbolic space, where we leverage the time-like dimension as a natural axis to capture a model's confidence and evolution. Building on this framework, we introduce two uncertainty-aware losses, robust hyperbolic contrastive loss and entailment cone loss, designed to enforce alignment and a partial-order relationship between the old and new models while considering the quality of the old embeddings. The next

---

[1] For simplicity, we abuse notation by letting $\phi$ to take a set of items $\mathcal{G}$ as the input and produce a vector database containing all embedded items.

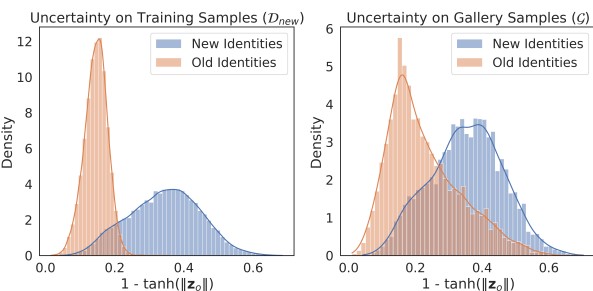

*Figure 2.* The distribution of uncertainty measures on hyperbolic embeddings of CIFAR100 produced by the old model $\phi_o$. The old model is ResNet18 trained with the first 50 classes (old identities) and then used to embed the upcoming 50 classes (new identities) into the vector database. The gallery is unseen samples from the test set. The curvature parameter $K$ is set to 1.0.

section presents our base hyperbolic encoder with an uncertainty measure for its embeddings. Section 4.2 and 4.3 will discuss the entailment and contrastive losses, respectively.

### 4.1. Hyperbolic Encoder and Uncertainty Estimation

We adopt a *hybrid* Euclidean–Hyperbolic model (Khrulkov et al., 2020) to encode images into hyperbolic space. We choose this hybrid model instead of a fully hyperbolic model (Bdeir et al., 2023) because it better integrates with any existing Euclidean encoders.

Let $\mathbf{z}$ be the vector embedding of the input $x$ produced by a given encoder (*e.g.,* ResNet (He et al., 2016) or ViT (Dosovitskiy, 2020)). We apply the exponential map $\mathrm{expm}_{\bar{\mathbf{0}}}$ to lift $[0, \mathbf{z}] \in \mathbb{R}^{d+1}$ from the tangent space at the hyperbolic origin $\bar{\mathbf{0}}$ onto the hyperboloid $\mathcal{L}^d$.

$$\phi^{\mathcal{L}}(x) = \mathrm{expm}_{\bar{\mathbf{0}}}([0, \mathbf{z}]^{\top}).$$

Here, $[0, \mathbf{z}] \in \mathbb{R}^{d+1}$ lies in the tangent space at the origin of the hyperboloid.

For convenience, in what follows, we denote the mappings from the input $x$ to the hyperbolic embedding for the old and new models as:

$$\phi_o^{\mathcal{L}}(x) : x \xrightarrow{\phi_o} \mathbf{z}_o \in \mathbb{R}^d \xrightarrow{\mathrm{expm}_{\bar{\mathbf{0}}}} \mathbf{h}_o \in \mathcal{L}^d$$

$$\phi_n^{\mathcal{L}}(x) : x \xrightarrow{\phi_n} \mathbf{z}_n \in \mathbb{R}^d \xrightarrow{\mathrm{expm}_{\bar{\mathbf{0}}}} \mathbf{h}_n \in \mathcal{L}^d$$

**Training Hyperbolic Encoder.** Similar to (Shen et al., 2020), we train the hyperbolic encoder as a classification task in hyperbolic space by using hyperbolic multinomial logistic regression (Hyperbolic MLR) (Bdeir et al., 2023). The probability of a class $y \in \mathcal{Y}$ given the embedding $\mathbf{h}$ can be expressed as:

$$p(Y = y|\mathbf{h}) \propto \exp\left(\mathrm{sign}(\langle \mathbf{w}_y, \mathbf{h} \rangle_{\mathcal{L}}) \|\mathbf{w}_y\|_{\mathcal{L}} d_{\mathcal{L}}(\mathbf{h}, \mathcal{H}_{\mathbf{w}_y})\right),$$
$$(9)$$

where $\mathcal{H}_{\mathbf{w}_y} = \{\mathbf{h} \in \mathcal{L}^d | \langle \mathbf{w}_y, \mathbf{h} \rangle = 0\}$ is the decision hyperplane of the class $y$ parametrized by $\mathbf{w}_y \in \mathcal{T}_{\bar{\mathbf{0}}}\mathcal{L}^d$. The term $d_{\mathcal{L}}(\mathbf{h}, \mathcal{H}_{\mathbf{w}_y})$ is the minimum distance from $\mathbf{h}$ to the hyperplane, which also admits a closed form using Euclidean reparameterization trick (Mishne et al., 2023). Since the base training is not our focus, we refer readers to (Bdeir et al., 2023) for full discussion. Our base objective is:

$$\mathcal{L}_{\mathrm{base}}(\mathbf{h}) = -\log p(Y = y|\mathbf{h}).$$

**Uncertainty Estimation.** Existing works (Franco et al., 2023; Atigh et al., 2022) typically use the hyperbolic uncertainty estimation in the Poincaré model, defining uncertainty as the $\ell_2$-norm of hyperbolic embeddings. We adapt this definition of uncertainty to the Lorentz model by leveraging its isometric equivalence with the Poincaré model:

$$\mathrm{Uncertainty}(\mathbf{h}) = 1 - \left\| \frac{\mathbf{h}_{\mathrm{space}}}{h_{time}} \right\|$$

$$= 1 - \frac{1}{\cosh(\sqrt{K}\|\mathbf{z}\|)} \left\| \frac{\sinh(\sqrt{K}\|\mathbf{z}\|)}{\sqrt{K}\|\mathbf{z}\|} \mathbf{z} \right\|$$

$$= 1 - \frac{1}{\sqrt{K}} \tanh\left(\sqrt{K}\|\mathbf{z}\|\right), \quad (10)$$

where $\mathbf{z}$ is the Euclidean embedding of $\mathbf{h}$ before the exponential map. This choice of the uncertainty measure results in the same formulation as the Pointcaré uncertainty measure used in Franco et al. (2023) and Atigh et al. (2022). For $K = 1$, the uncertainty is bounded in the range $[0, 1]$.

Figure 2 shows the distribution of the hyperbolic uncertainty estimates for embeddings produced by ResNet18 trained on the first 50 classes of CIFAR100. When this *old* model is used to embed images from the *new* 50 classes (unseen during its training), the resulting embeddings exhibit higher uncertainty and lower norms $\|\mathbf{z}_o\|$ compared to images from the first 50 classes (seen during the training). This observation is consistent with (Atigh et al., 2022), showing the close relationship between an embedding's norm and its associated uncertainty in hyperbolic models.

### 4.2. Entailment Models Evolution

When training the new model, we aim to keep it consistent with the structure discovered by the old model and then refine previously learned relationships. We model this consistency-then-refinement dynamic via the notion of *entailment cone* (Desai et al., 2023; Ganea et al., 2018), which constrains the region where new embeddings can 'descend' from old embeddings in hyperbolic space. In other words, the cone represents the set of permissible embedding adjustments as the model evolves.

Formally, the *entailment cone* at $\mathbf{h}_o$ in the hyperbolic space is defined by a *half-aperture* function that diminishes with

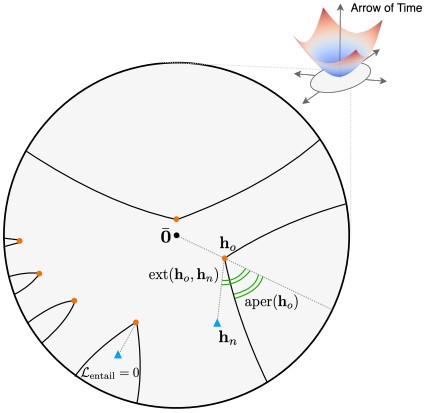

*Figure 3.* Illustration of the entailment cone. The more certain $\mathbf{h}_o$, the narrower the cone defined by $\mathbf{h}_o$.

distance from the hyperbolic origin (Desai et al., 2023):

$$\text{aper}(\mathbf{h}_o) = \sin^{-1}\left(\frac{2\epsilon}{\sqrt{K}\,\|\mathbf{h}_{o,\text{space}}\|}\right), \qquad (11)$$

where $\epsilon$ is a small constant (e.g. 0.1) preventing the degenerate behavior near the origin.

The hyperbolic entailment cone loss penalizes the cases when $\mathbf{h}_n$ lies outside $\mathbf{h}_o$'s entailment cone using the differences between the *exterior angle* $\text{ext}(\mathbf{h}_o, \mathbf{h}_n) = \pi - \angle(\mathbf{O}, \mathbf{h}_o, \mathbf{h}_n)$ and the aperture (see Figure 3):

$$\mathcal{L}_{\text{entail}}(\mathbf{h}_n; \mathbf{h}_o) = \max\left(0,\ \text{ext}(\mathbf{h}_o, \mathbf{h}_n) - \text{aper}(\mathbf{h}_o)\right),$$

where the exterior angle is computed by

$$\text{ext}(\mathbf{h}_o, \mathbf{h}_n) = \cos^{-1}\left(\frac{h_{n,\text{time}} + h_{o,\text{time}}\left(K\langle\mathbf{h}_o,\mathbf{h}_n\rangle_\mathcal{L}\right)}{\|\mathbf{h}_{o,\text{space}}\|\sqrt{\left(K\langle\mathbf{h}_o,\mathbf{h}_n\rangle_\mathcal{L}\right)^2 - 1}}\right).$$

Here, we penalize $\mathbf{h}_n$ for each pair $(\mathbf{h}_o, \mathbf{h}_n)$ if it lies outside of the entailment cone dictated by $\mathbf{h}_o$. We refer readers to (Desai et al., 2023) for full derivations.

It is worth noting from Eq. (11) that the aperture at $\mathbf{h}_o$ is inversely proportional to its norm. Therefore, as $\mathbf{h}_o$ becomes more uncertain (lower norm), its cone 'widens', granting greater freedom for the evolutionary expansion in the new embedding space.

### 4.3. Uncertainty-aware Contrastive Alignment

Notice that the entailment cone restricts only the spatial coordinate changes of the embedding, without preserving pairwise distance relationships between old and new embeddings. Meanwhile, contrastive alignment loss (Biondi et al., 2024; Zhang et al., 2022a) or mean distortion (Cui et al., 2024) may hinder the learning of the new model since they naively enforce the new model to reconstruct outdated representations regardless of their quality. This may exacerbate

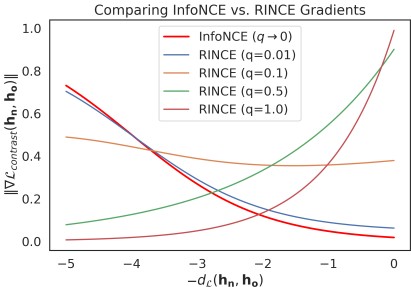

*Figure 4.* Comparison between InfoNCE and RINCE losses for varying distances between old and new embedding $\mathbf{h}_o$ and $\mathbf{h}_n$. As $q$ increases, the gradient norm decreases for noisy positive pairs ($-d_\mathcal{L} \in [-5, -3]$) and increases for clean positive pairs ($-d_\mathcal{L} \in [-2, -0]$). For this illustration, we choose one negative sample with the distance $d_\mathcal{L}(\mathbf{h}_n, \mathbf{h}'_o) = 4.0$ and $\beta = 0.01$.

the inherent trade-off between maintaining compatibility and achieving high performance in the new model.

To address this issue, we propose a variant of RINCE–robust contrastive loss (Chuang et al., 2022)–that adaptively adjusts the weight for *positive* old-new embedding pairs according to the uncertainty of the old embeddings.

Let $\mathcal{B}$ be a batch of images in the training data, RINCE loss for a pair $(\mathbf{h}_o, \mathbf{h}_n) \in \mathcal{B}$ is given by

$$\mathcal{L}_{contrast}(\mathbf{h}_n; \beta, q) = -\frac{1}{q}\exp(-qd_\mathcal{L}(\mathbf{h}_n, \mathbf{h}_o))$$

$$+ \frac{1}{q}\left(\beta\sum_{\mathbf{h}'_o \in \mathcal{B}}\exp(-d_\mathcal{L}(\mathbf{h}_n, \mathbf{h}'_o))\right)^q, \quad (12)$$

where $\beta, q \in (0, 1]$ are hyper-parameters of the RINCE loss. As $q \to 0$, RINCE recovers vanilla InfoNCE (Oord et al., 2018) loss. As $q \to 1$, RINCE reduces the weights for noisy pairs (with large distances) and focuses more on clean pairs (low distances)–see Figure 4 for the comparison of the gradient norms for InfoNCE and RINCE with varying $q$.

The RINCE loss is particularly suitable in the context of compatibility, where the goal is to emphasize the reconstruction of *good* old embeddings while downweighting *bad* ones. To achieve this, we make an explicit dependence of $q$ to $\mathbf{h}_o$ for each pair $(\mathbf{h}_n, \mathbf{h}_o)$, defining $q(\mathbf{h}_o) = \text{Uncertainty}(\mathbf{h}_o)$. This allows the weights for positive pairs to adjust dynamically based on the quality of the old embeddings.

### 4.4. Training Pipeline

The overall loss function to train the updated model is:

$$\mathcal{L} = \mathcal{L}_{\text{base}} + \lambda(\mathcal{L}_{\text{entail}} + \mathcal{L}_{\text{contrast}}), \qquad (13)$$

where $\lambda$ is the weight for the alignment regularization. Here, we make no assumption about $\mathcal{L}_{\text{base}}$, making it applicable to various training scenarios, including unsupervised and

self-supervised learning, which drive recent advancements in foundation models. We leave further exploration of these settings for future work.

**Numerical Stability.** Because the hyperbolic distance grows quickly with respect to the norm $\|\mathbf{z}\|$, it is often necessary to *clip* or *rescale* norms to maintain well-conditioned embeddings. We follow (Bdeir et al., 2023) and (Desai et al., 2023) to rescale the norm of $\mathbf{z}$ by $\sqrt{d}$ and then clipping at $\zeta$ to ensure stability. For the old model, we set $\zeta_o = 1.0$ and for the new model, we increase the clipping threshold by $\zeta_n = \zeta_o + 0.2$ to account for the evolution in the embedding space of the updated model.

## 5. Experiments

We conduct extensive experiments and compare with existing approaches in Euclidean space to demonstrate the effectiveness of our method. We refer to our method as *Hyperbolic Backward Compatible Training (HBCT)*. The source code is available at `https://github.com/snap-research/hyperbolic_bct`.

### 5.1. Experimental Settings

**Datasets.** We use CIFAR100 (Krizhevsky et al., 2009) and Tiny-Imagenet (Le & Yang, 2015), where the test sets serve as hold-out query and gallery sets for evaluating both retrieval performance and compatibility.

**Models.** We employ two model architectures: ResNet18 (He et al., 2016) and ViT-B-16 (Dosovitskiy, 2020). ResNet18 is trained from scratch, whereas ViT-B-16 is initialized with weights pre-trained on ImageNet21k (Ridnik et al., 2021). The goal is to replicate the typical practice of obtaining and fine-tuning pre-trained models, which then function as external alternatives to replace the currently fine-tuned model.

**Scenarios.** We consider four settings to capture a range of real-world use cases:

- *Extended data*: We randomly select $30\%$ of the training dataset and use it as $\mathcal{D}_{old}$ to train the old model. $\mathcal{D}_{new}$ will be the the full dataset. The model architecture is ResNet18 and is kept the same for the new model.

- *Extended class*: We extract from the first $50\%$ of classes ($\sim 50$ classes for CIFAR100 and $\sim 100$ classes for TinyImagenet) as $\mathcal{D}_{old}$. Meanwhile, $\mathcal{D}_{new}$ will be the full dataset. The model architecture is ResNet18 and is kept the same for the new model.

- *New architecture*: We assume that we have access to the full training dataset, *i.e.*, $\mathcal{D}_{old} = \mathcal{D}_{new}$. The old model is ResNet18, and the new model finetunes on ViT-B-16 pretrained on Imagenet21k.

- *Both*: We simulate the arrival of new classes and a new architecture. The old model is ResNet18 trained with $50\%$ of classes, and the new model is ViT-B-16 finetuned on the full dataset.

**Baselines.** We compare our method with other regularization-based methods in Euclidean space, including $\ell_2$ distance regularization, *BCT* (Shen et al., 2020), *Hot-Refresh*[2] (Zhang et al., 2022a), *AdvBCT* (Pan et al., 2023), and *HOC*[3] (Biondi et al., 2024).

**Metrics.** We assess retrieval performance using two metrics: (1) *Cumulative Matching Characteristics (CMC@k)*: This metric measures top-$k$ accuracy by ranking gallery representations based on their similarity to the query representation. A query is considered correctly retrieved if a sample from the same class appears within the top-$k$ ranked results. We report CMC top-1 and top-5 scores for all models; (2) *Mean Average Precision (mAP)*: mAP integrates precision and recall by computing the area under the precision-recall curve. We use mAP@1.0, which represents the average precision over recall values ranging from 0.0 to 1.0.

We define metrics to assess the backward-compatibility of an alignment method following (Shen et al., 2020; Pan et al., 2023). Let $\phi_*$ be the base new model trained without any alignment method. The compatibility metric for a new model $\phi_n$ trained with an alignment method is computed by

$$\mathcal{P}_{\text{com}} = \frac{\mathcal{M}(\phi_n(\mathcal{Q}); \phi_o(\mathcal{G})) - \mathcal{M}(\phi_o(\mathcal{Q}); \phi_o(\mathcal{G}))}{\mathcal{M}(\phi_*(\mathcal{Q}); \phi_*(\mathcal{G})) - \mathcal{M}(\phi_o(\mathcal{Q}); \phi_o(\mathcal{G}))}, \quad (14)$$

where $\mathcal{M}(\cdot ; \cdot)$ is an evaluation metric for the retrieval task, which is either CMC@k or mAP. Since backward-compatibility may come with a trade-off in the performance of the new model, we also compare the performance with the base model without alignment $\phi_*$

$$\mathcal{P}_{\text{up}} = \frac{\mathcal{M}(\phi_n(\mathcal{Q}); \phi_o(\mathcal{G})) - \mathcal{M}(\phi_*(\mathcal{Q}); \phi_*(\mathcal{G}))}{\mathcal{M}(\phi_*(\mathcal{Q}); \phi_*(\mathcal{G}))}. \quad (15)$$

**Implementation Details.** For all the models, we use the feature dimension of 128. For ResNet18 model, we use 200 epochs, learning rate 0.1 and for finetuning ViT-B-16, we use 80 epochs with learning rate 0.005. For both models, we use SGD (Ruder, 2016) as the optimizer with a batch size of

---

[2]Hot-Refresh and HOC are two alignment methods that leverage hyperspherical geometry by using a contrastive objective based on cosine similarity. The key difference between them lies in the selection of negative pairs.

[3]Note that in (Biondi et al., 2024), the authors proposed using a fixed $d$-simplex classifier to improve compatibility. However, this approach requires predefining the number of dimensions to match the number of classes and maintaining it consistently across model updates, which is usually not practical. Therefore, we adopt only their high-order alignment loss (HOC) while removing the constraint of a fixed classifier.

*Table 1.* The comparisons of different backward-compatibility methods. Here, $\phi^{\mathbb{R}}$ indicates that the model produces Euclidean embeddings while $\phi^{\mathcal{L}}$ produces hyperbolic embeddings. The columns *self* and *cross* refer to the original new-to-new and new-to-old retrieval performance (CMC@1 or mAP), respectively. Gray rows are base models without compatibility, and the pink rows indicate the proposed method. **Bold** indicates the best performance and underline indicates the second best method. When the ablated version of HBCT (*i.e.*, without entailment loss) outperforms the best Euclidean baseline, we highlight it by a dashed underline.

| Scenario | Old Model | New Model | CIFAR100 CMC@1 self | cross | $\mathcal{P}_{up}$ | $\mathcal{P}_{com}$ | CIFAR100 mAP self | cross | $\mathcal{P}_{up}$ | $\mathcal{P}_{com}$ | TinyImageNet CMC@1 self | cross | $\mathcal{P}_{up}$ | $\mathcal{P}_{com}$ | TinyImageNet mAP self | cross | $\mathcal{P}_{up}$ | $\mathcal{P}_{com}$ |
|---|---|---|---|---|---|---|---|---|---|---|---|---|---|---|---|---|---|---|
| Ext-data | $\phi_o^{\mathbb{R}}$ | – | 0.572 | – | – | – | 0.472 | – | – | – | 0.373 | – | – | – | 0.248 | – | – | – |
| | $\phi_o^{\mathbb{R}}$ | $\phi_*^{\mathbb{R}}$ | 0.716 | 0.019 | 0.000 | -3.846 | 0.654 | 0.072 | 0.000 | -2.195 | 0.540 | 0.005 | 0.000 | -2.200 | 0.420 | 0.010 | 0.000 | -1.384 |
| | $\phi_o^{\mathbb{R}}$ | $\phi_n^{\mathbb{R}}$-$\ell_2$ | 0.680 | 0.645 | -0.049 | 0.510 | 0.627 | 0.537 | -0.042 | 0.355 | 0.538 | 0.474 | -0.004 | 0.603 | 0.421 | 0.311 | 0.002 | 0.362 |
| | $\phi_o^{\mathbb{R}}$ | $\phi_n^{\mathbb{R}}$-BCT | 0.706 | 0.701 | -0.014 | 0.900 | 0.642 | 0.546 | -0.018 | 0.406 | 0.527 | 0.490 | -0.024 | 0.702 | 0.407 | 0.309 | -0.032 | 0.354 |
| | $\phi_o^{\mathbb{R}}$ | $\phi_n^{\mathbb{R}}$-Hot-Refresh | 0.718 | 0.671 | 0.003 | 0.693 | 0.653 | 0.544 | -0.003 | 0.396 | 0.545 | 0.475 | 0.009 | 0.610 | 0.425 | 0.312 | 0.011 | 0.369 |
| | $\phi_o^{\mathbb{R}}$ | $\phi_n^{\mathbb{R}}$-AdvBCT | 0.698 | 0.681 | -0.025 | 0.760 | 0.638 | 0.543 | -0.025 | 0.389 | 0.493 | 0.421 | -0.088 | 0.286 | 0.376 | 0.287 | -0.105 | 0.227 |
| | $\phi_o^{\mathbb{R}}$ | $\phi_n^{\mathbb{R}}$-HOC | 0.720 | 0.670 | 0.006 | 0.685 | 0.656 | 0.544 | 0.003 | 0.396 | 0.552 | 0.475 | 0.022 | 0.610 | 0.428 | 0.313 | 0.019 | 0.377 |
| | $\phi_o^{\mathcal{L}}$ | – | 0.576 | – | – | – | 0.455 | – | – | – | 0.379 | – | – | – | 0.233 | – | – | – |
| | $\phi_o^{\mathcal{L}}$ | $\phi_*^{\mathcal{L}}$ | 0.722 | 0.011 | 0.000 | -3.851 | 0.651 | 0.012 | 0.000 | -2.263 | 0.548 | 0.002 | 0.000 | -2.230 | 0.410 | 0.006 | 0.000 | -1.282 |
| | $\phi_o^{\mathcal{L}}$ | $\phi_n^{\mathcal{L}}$-HBCT w/o entail | 0.722 | 0.700 | -0.001 | 0.847 | 0.653 | 0.564 | -0.002 | 0.547 | 0.542 | 0.542 | -0.011 | 0.966 | 0.409 | 0.329 | -0.002 | 0.539 |
| | $\phi_o^{\mathcal{L}}$ | $\phi_n^{\mathcal{L}}$-HBCT | 0.721 | 0.716 | -0.002 | **0.956** | 0.656 | 0.567 | 0.001 | **0.560** | 0.551 | 0.560 | 0.007 | **1.076** | 0.414 | 0.334 | 0.009 | **0.572** |
| Ext-class | $\phi_o^{\mathbb{R}}$ | – | 0.376 | – | – | – | 0.310 | – | – | – | 0.273 | – | – | – | 0.211 | – | – | – |
| | $\phi_o^{\mathbb{R}}$ | $\phi_*^{\mathbb{R}}$ | 0.716 | 0.019 | 0.000 | -1.052 | 0.654 | 0.019 | 0.000 | -0.845 | 0.540 | 0.007 | 0.000 | -0.991 | 0.420 | 0.008 | 0.000 | -0.971 |
| | $\phi_o^{\mathbb{R}}$ | $\phi_n^{\mathbb{R}}$-$\ell_2$ | 0.712 | 0.478 | -0.005 | 0.302 | 0.653 | 0.342 | -0.002 | 0.093 | 0.536 | 0.307 | -0.008 | 0.127 | 0.418 | 0.226 | -0.005 | 0.071 |
| | $\phi_o^{\mathbb{R}}$ | $\phi_n^{\mathbb{R}}$-BCT | 0.695 | 0.447 | -0.029 | 0.210 | 0.624 | 0.328 | -0.046 | 0.054 | 0.514 | 0.292 | -0.049 | 0.073 | 0.401 | 0.217 | -0.046 | 0.029 |
| | $\phi_o^{\mathbb{R}}$ | $\phi_n^{\mathbb{R}}$-Hot-Refresh | 0.715 | 0.498 | -0.001 | 0.360 | 0.649 | 0.348 | -0.007 | 0.110 | 0.534 | 0.334 | -0.012 | 0.229 | 0.414 | 0.231 | -0.016 | 0.094 |
| | $\phi_o^{\mathbb{R}}$ | $\phi_n^{\mathbb{R}}$-AdvBCT | 0.708 | 0.459 | -0.011 | 0.244 | 0.642 | 0.337 | -0.018 | 0.080 | 0.524 | 0.254 | -0.030 | -0.070 | 0.404 | 0.213 | -0.039 | 0.007 |
| | $\phi_o^{\mathbb{R}}$ | $\phi_n^{\mathbb{R}}$-HOC | 0.713 | 0.490 | -0.003 | 0.336 | 0.651 | 0.346 | -0.005 | 0.106 | 0.537 | 0.342 | -0.005 | 0.258 | 0.416 | 0.233 | -0.011 | 0.102 |
| | $\phi_o^{\mathcal{L}}$ | – | 0.425 | – | – | – | 0.330 | – | – | – | 0.299 | – | – | – | 0.212 | – | – | – |
| | $\phi_o^{\mathcal{L}}$ | $\phi_*^{\mathcal{L}}$ | 0.722 | 0.002 | 0.000 | -1.428 | 0.656 | 0.012 | 0.000 | -0.979 | 0.549 | 0.006 | 0.000 | -1.178 | 0.410 | 0.007 | 0.000 | -1.038 |
| | $\phi_o^{\mathcal{L}}$ | $\phi_n^{\mathcal{L}}$-HBCT w/o entail | 0.726 | 0.555 | 0.006 | 0.435 | 0.655 | 0.388 | -0.001 | 0.176 | 0.547 | 0.344 | -0.003 | 0.177 | 0.404 | 0.236 | -0.014 | 0.120 |
| | $\phi_o^{\mathcal{L}}$ | $\phi_n^{\mathcal{L}}$-HBCT | 0.722 | 0.572 | -0.001 | **0.495** | 0.655 | 0.398 | -0.001 | **0.209** | 0.553 | 0.373 | 0.009 | **0.295** | 0.408 | 0.246 | -0.005 | **0.169** |
| New-Arch | $\phi_o^{\mathbb{R}}$ | – | 0.714 | – | – | – | 0.652 | – | – | – | 0.545 | – | – | – | 0.423 | – | – | – |
| | $\phi_o^{\mathbb{R}}$ | $\phi_*^{\mathbb{R}}$ | 0.900 | 0.021 | 0.000 | -3.706 | 0.879 | 0.029 | 0.000 | -2.744 | 0.812 | 0.002 | 0.000 | -2.039 | 0.745 | 0.006 | 0.000 | -1.292 |
| | $\phi_o^{\mathbb{R}}$ | $\phi_n^{\mathbb{R}}$-$\ell_2$ | 0.896 | 0.778 | -0.005 | 0.343 | 0.873 | 0.728 | -0.006 | 0.337 | 0.807 | 0.703 | -0.006 | 0.593 | 0.732 | 0.534 | -0.017 | 0.346 |
| | $\phi_o^{\mathbb{R}}$ | $\phi_n^{\mathbb{R}}$-BCT | 0.897 | 0.830 | -0.004 | 0.623 | 0.882 | 0.750 | 0.003 | 0.432 | 0.809 | 0.757 | -0.003 | 0.794 | 0.738 | 0.546 | -0.009 | 0.384 |
| | $\phi_o^{\mathbb{R}}$ | $\phi_n^{\mathbb{R}}$-Hot-Refresh | 0.897 | 0.810 | -0.003 | 0.517 | 0.876 | 0.743 | -0.003 | 0.402 | 0.804 | 0.720 | -0.009 | 0.656 | 0.731 | 0.537 | -0.019 | 0.356 |
| | $\phi_o^{\mathbb{R}}$ | $\phi_n^{\mathbb{R}}$-AdvBCT | 0.891 | 0.797 | -0.011 | 0.449 | 0.868 | 0.737 | -0.012 | 0.374 | 0.799 | 0.658 | -0.016 | 0.423 | 0.722 | 0.517 | -0.031 | 0.294 |
| | $\phi_o^{\mathbb{R}}$ | $\phi_n^{\mathbb{R}}$-HOC | 0.895 | 0.813 | -0.006 | 0.534 | 0.877 | 0.744 | -0.002 | 0.408 | 0.804 | 0.720 | -0.009 | 0.655 | 0.728 | 0.537 | -0.023 | 0.354 |
| | $\phi_o^{\mathcal{L}}$ | – | 0.723 | – | – | – | 0.657 | – | – | – | 0.546 | – | – | – | 0.413 | – | – | – |
| | $\phi_o^{\mathcal{L}}$ | $\phi_*^{\mathcal{L}}$ | 0.897 | 0.009 | 0.000 | -4.085 | 0.882 | 0.011 | 0.000 | -2.876 | 0.809 | 0.002 | 0.000 | -2.069 | 0.746 | 0.006 | 0.000 | -1.221 |
| | $\phi_o^{\mathcal{L}}$ | $\phi_n^{\mathcal{L}}$-HBCT w/o entail | 0.898 | 0.6915 | 0.001 | -0.178 | 0.882 | 0.680 | -0.001 | 0.099 | 0.8137 | 0.7936 | 0.006 | 0.941 | 0.751 | 0.571 | 0.006 | 0.475 |
| | $\phi_o^{\mathcal{L}}$ | $\phi_n^{\mathcal{L}}$-HBCT | 0.893 | 0.886 | -0.004 | **0.937** | 0.879 | 0.782 | -0.004 | **0.555** | 0.811 | 0.804 | 0.002 | **0.982** | 0.746 | 0.580 | 0.000 | **0.500** |
| Both | $\phi_o^{\mathbb{R}}$ | – | 0.386 | – | – | – | 0.315 | – | – | – | 0.281 | – | – | – | 0.222 | – | – | – |
| | $\phi_o^{\mathbb{R}}$ | $\phi_*^{\mathbb{R}}$ | 0.900 | 0.018 | 0.000 | -0.716 | 0.879 | 0.028 | 0.000 | -0.509 | 0.811 | 0.007 | 0.000 | -0.529 | 0.744 | 0.005 | 0.000 | -0.425 |
| | $\phi_o^{\mathbb{R}}$ | $\phi_n^{\mathbb{R}}$-$\ell_2$ | 0.888 | 0.482 | -0.013 | 0.185 | 0.857 | 0.382 | -0.025 | 0.119 | 0.794 | 0.451 | -0.021 | 0.321 | 0.712 | 0.293 | -0.042 | 0.137 |
| | $\phi_o^{\mathbb{R}}$ | $\phi_n^{\mathbb{R}}$-BCT | 0.895 | 0.358 | -0.006 | -0.054 | 0.880 | 0.345 | 0.002 | 0.053 | 0.818 | 0.404 | 0.009 | 0.233 | 0.750 | 0.282 | 0.008 | 0.115 |
| | $\phi_o^{\mathbb{R}}$ | $\phi_n^{\mathbb{R}}$-Hot-Refresh | 0.891 | 0.490 | -0.010 | 0.202 | 0.867 | 0.384 | -0.013 | 0.122 | 0.802 | 0.474 | -0.011 | **0.365** | 0.732 | 0.296 | -0.015 | 0.143 |
| | $\phi_o^{\mathbb{R}}$ | $\phi_n^{\mathbb{R}}$-AdvBCT | 0.878 | 0.513 | -0.025 | 0.245 | 0.842 | 0.377 | -0.041 | 0.111 | 0.802 | 0.416 | -0.010 | 0.255 | 0.730 | 0.288 | -0.018 | 0.126 |
| | $\phi_o^{\mathbb{R}}$ | $\phi_n^{\mathbb{R}}$-HOC | 0.896 | 0.446 | -0.004 | 0.116 | 0.869 | 0.377 | -0.010 | 0.110 | 0.798 | 0.412 | -0.016 | 0.247 | 0.725 | 0.287 | -0.025 | 0.126 |
| | $\phi_o^{\mathcal{L}}$ | – | 0.420 | – | – | – | 0.332 | – | – | – | 0.296 | – | – | – | 0.213 | – | – | – |
| | $\phi_o^{\mathcal{L}}$ | $\phi_*^{\mathcal{L}}$ | 0.896 | 0.025 | 0.000 | -0.831 | 0.881 | 0.013 | 0.000 | -0.583 | 0.814 | 0.008 | 0.000 | -0.556 | 0.753 | 0.006 | 0.000 | -0.384 |
| | $\phi_o^{\mathcal{L}}$ | $\phi_n^{\mathcal{L}}$-HBCT w/o entail | 0.897 | 0.188 | 0.000 | -0.487 | 0.881 | 0.221 | 0.000 | -0.204 | 0.817 | 0.192 | 0.003 | -0.201 | 0.752 | 0.183 | -0.002 | -0.057 |
| | $\phi_o^{\mathcal{L}}$ | $\phi_n^{\mathcal{L}}$-HBCT | 0.897 | 0.575 | 0.001 | **0.324** | 0.879 | 0.417 | -0.002 | **0.154** | 0.812 | 0.476 | -0.003 | 0.348 | 0.745 | 0.313 | -0.011 | **0.184** |

128. We set the weight decay to 5e-4 and momentum to 0.9, and use a cosine annealing schedule to tune the learning rate. The old models are trained only with the base classification loss in Eq. (8), and the new model is equipped with one of the alignment methods. We vary the alignment weight $\lambda$ in the range $[0, 1.0]$ and for methods involve contrastive alignment loss, we choose the temperature $\tau \in \{0.5, 1.0\}$. For each baseline, we run with 10 combinations of these hyperparameters and choose the best run that at least does not hurt the performance of the new model. For our method, we fix the curvature $K = 1.0$ and we found that performance is consistent across runs, so we fix $\lambda = 0.3, \tau = 0.5, \beta = 0.01$ (similar to RINCE) for all the settings. The retrieval distance used for Euclidean models is the cosine similarity distance while Lorentz models is the geodesic distance as in Eq. (3).

*Table 2.* Ablating objectives in HBCT.

| Ablation | CMC@1 $\mathcal{P}_{up}$ | $\mathcal{P}_{com}$ | mAP $\mathcal{P}_{up}$ | $\mathcal{P}_{com}$ |
|---|---|---|---|---|
| $\phi_n^{\mathcal{L}}$-Hot-Refresh | -0.001 | 0.36 | -0.007 | 0.110 |
| $\phi_n^{\mathcal{L}}$-HBCT w/ RINCE w/ entail | -0.001 | 0.495 | -0.001 | 0.209 |
| $\phi_n^{\mathcal{L}}$-HBCT w/ RINCE w/o entail | 0.006 | 0.435 | -0.001 | 0.176 |
| $\phi_n^{\mathcal{L}}$-HBCT w/ InfoNCE w/ entail | -0.002 | 0.475 | -0.009 | 0.202 |
| $\phi_n^{\mathcal{L}}$-HBCT w/ InfoNCE w/o entail | 0.001 | 0.431 | -0.001 | 0.193 |

## 5.2. Empirical Results

Table 1 reports the compatibility performance of competing methods. Notably, HBCT's $\mathcal{P}_{com}$ surpasses other Euclidean baselines by a substantial margin in most settings. In particular, HBCT improves CMC@1 by 21.4% on average compared to the strongest Euclidean baseline, and yields a 44.8% increase in mAP. These compatibility gains can be achieved

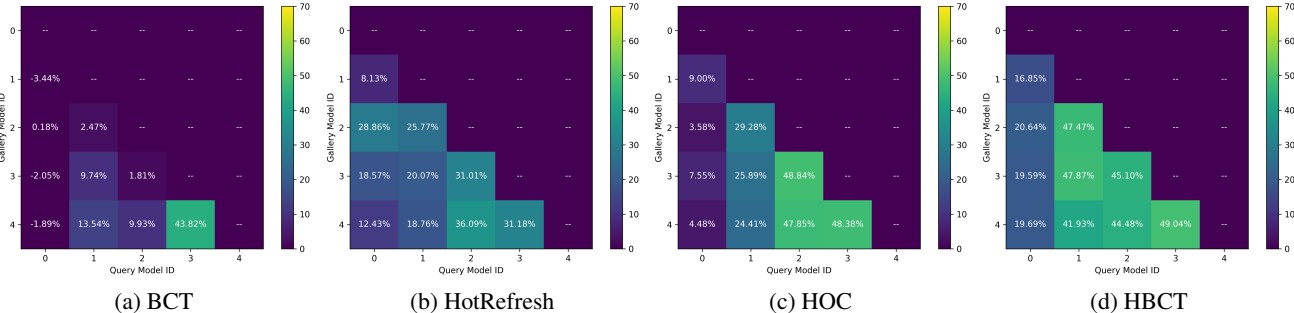

(a) BCT            (b) HotRefresh            (c) HOC            (d) HBCT

*Figure 5.* The compatibility matrix of different alignment methods after five consecutive model updates. The value at column $j$ and row $i$ quantifies the CMC@1 compatibility $\mathcal{P}_{\text{com}}$ where the queries are embedded by the model $\phi_i$ and the gallery is embedded by $\phi_j$.

with only minimal compromises to the new model's performance, as shown by the $\mathcal{P}_{\text{up}}$ scores. Meanwhile, among alignment methods in Euclidean space, contrastive-based losses (HOC and Hot-Refresh) show relatively consistent performance. BCT achieves strong compatibility in certain scenarios (e.g., extended data and new architectures) but often entails a sacrifice in the new model's performance.

Note that we do not directly compare methods based on original retrieval metrics such as CMC@1 or mAP shown in *self* (new-to-new) and *cross* (new-to-old) columns in Table 1, because the 'old model' differs between Euclidean and hyperbolic spaces ($\phi_o^{\mathbb{R}}$ vs. $\phi_o^{\mathcal{L}}$), leading to inherently different retrieval performance. For instance, in extended-class scenarios, hyperbolic encoders often yield better old-model performance than their Euclidean counterparts, whereas Euclidean encoders generally excel in extended-data and new-architecture scenarios. This behavior may stem from hyperbolic representations' stronger ability to handle out-of-distribution samples (Li et al., 2024). However, this difference is not the main focus of our paper. Instead, we primarily rely on $\mathcal{P}_{\text{com}}$ and $\mathcal{P}_{\text{up}}$, which are calibrated to account for variations in old-model performance.

### 5.3. Sequential Model Updates

This section investigates the compatibility of alignment methods when applied over multiple model updates. We used the CIFAR-100 dataset for this purpose, segmenting the training data into five sets. Specifically, 20 new classes were incrementally added with each update, leading to stages with 20, 40, 60, 80, and ultimately all 100 classes. The base model for the initial two updates (time steps 0 and 1) is ResNet18, which is then transitioned to ResNet50 for the remaining three updates (time steps 2, 3, and 4) to explore architectural shifts. For each method, we report the CMC@1 compatibility matrix with $\mathcal{P}_{\text{com}}$ for HBCT, HOC, HotRefresh, and BCT. The results in Figure 5 show that HBCT provides superior compatibility maintenance with prior models across multiple updates, while the compatibil-

ity of baseline methods rapidly declines.

### 5.4. Ablation and Discussion

We also showed in Table 1 the performance of an ablated version of HBCT where we remove the entailment cone loss. We observe a significant drop in terms of backward compatibility in scenarios where we update the architecture to ViT-B-16. However, in some scenarios, HBCT, HBCT without entailment cone loss, also outperforms the best baselines in Euclidean space. We conduct an ablation study to further probe this behavior.

In the extended-class scenario on the CIFAR100 dataset, we replace the RINCE loss with a standard InfoNCE loss (using geodesic distance in hyperbolic space as the similarity measure) to evaluate its impact. In Table 2, we observe that RINCE offers a substantial improvement in CMC@1-based compatibility, while also reducing the sacrifice in mAP's $\mathcal{P}_{\text{up}}$. Interestingly, even HBCT with vanilla InfoNCE on the hyperbolic space can outperform the best baseline in Euclidean space. This aligns with our intuition that hyperbolic geometry better captures the evolution of the embedding space by organizing embeddings across models along the time dimension depending on their capabilities. Figure 6 in the appendix illustrates how the embeddings evolve from the old model to the new one. As the model improves, the uncertainty in the gallery embeddings decreases. Additional ablations for hyperparameters are provided in the appendix.

## 6. Related Work

**Compatible Representation Learning.** Aligning representations across deep learning models has been widely studied in applications like representation transfer (Li et al., 2015) and model stitching (Bansal et al., 2021) to multi-modal foundation models (Fei et al., 2022). In retrieval-based applications, Shen et al. (2020) is the first to introduce the concept of backward-compatible training to avoid backfilling the indexed embeddings. They enforce compatibility between

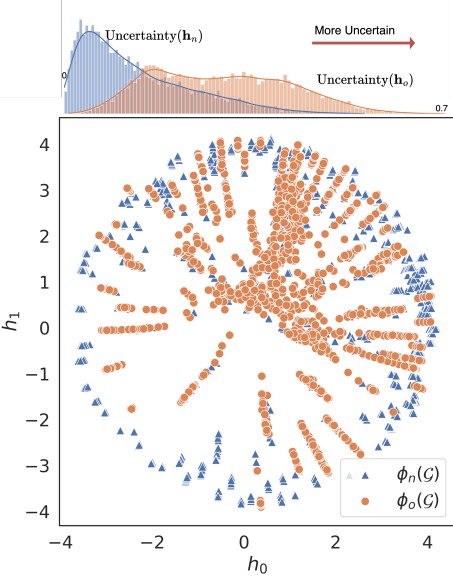

*Figure 6.* Visualization of old and new gallery embeddings in CIFAR100. We compress 128-dimensional embeddings into a 2-dimensional hyperboloid using UMAP (McInnes et al., 2018) and visualize them in the tangent space $\mathcal{T}_0\mathcal{L}^2$. The top histogram is the distribution of the uncertainty estimation for those embeddings.

old and new models by transferring the learned classifier from the old model to influence the new one. While this method facilitates compatibility, it can compromise the expressiveness of the new model. To address this, subsequent studies have explored various techniques to improve the regularization loss, such as contrastive-based alignment(Zhang et al., 2022a; Biondi et al., 2024), adversarial loss (Pan et al., 2023), and the alignment of class centroids between old and new models (Zhang et al., 2022b). However, these approaches primarily rely on Euclidean geometry, while we shift the focus to hyperbolic geometry and leverage its natural uncertainty estimation to adaptively adjust the alignment.

On a different front, Hu et al. (2022); Wang et al. (2020) proposed adding an extra projection to map the old embedding into the new embedding space, while Zhou et al. (2023); Ramanujan et al. (2022) introduced auxiliary dimensions to increase the expressiveness of either the old or new embeddings. Building on backward-compatible models, recent research has explored dynamic techniques to progressively backfill the gallery in real-time (Zhang et al., 2022a; Jaeckle et al., 2023; Yan et al., 2021). These approaches are orthogonal and can be integrated into our method.

**Hyperbolic Representation Learning.** Hyperbolic geometry is primarily known for its ability to better capture hierarchical, tree-like structures (Yang et al., 2023; Peng et al., 2021), which enhances performance in various tasks, including molecular generation (Liu et al., 2019), recommendation (Yang et al., 2021; Li et al., 2021), image re-

trieval (Khrulkov et al., 2020; Wei et al., 2024), and information retrieval (Ganea et al., 2018; Dhingra et al., 2018). Recent studies have also demonstrated the benefits of hyperbolic geometry for (multi-modal) foundation models (Desai et al., 2023; Ibrahimi et al., 2024; Pal et al., 2024; Yang et al., 2024). In contrast to these works, we are the first to investigate the application of hyperbolic geometry in capturing the evolution of models across updates.

## 7. Conclusion

We introduced HBCT, a framework that leverages hyperbolic geometry to ensure seamless model updates while accounting for representation uncertainty. By constraining new embeddings within the entailment cone of old ones and introducing a dynamic hyperbolic contrastive loss, HBCT maintains compatibility without hindering expressiveness. Experiments show that our approach outperforms Euclidean-based methods in achieving robust backward compatibility. This work highlights the potential of hyperbolic geometry for scalable model evolution.

**Limitations and future work.** While HBCT is adaptable to various settings, we follow prior work by focusing on supervised learning scenarios. Investigating its effectiveness in self-supervised learning or multi-modal settings is an interesting research direction. Furthermore, to account for embedding space evolution, we gradually increase the clipping norm threshold. This may lead to instability as norms grow (*e.g.,* $\geq 10.$). A potential solution is to backfill and rescale the time dimension after a set number of updates (*e.g.,* 50 updates), though developing a more principled approach for handling this challenge is an important direction for future work.

## Acknowledgements

The authors gratefully acknowledge the generous support of Snap Research, the Amazon Research award, and the Samsung Research America award to the lab. Their funding and resources were instrumental in enabling and completing this work.

## Impact Statement

This paper presents work whose goal is to advance the field of Machine Learning. There are many potential societal consequences of our work, none which we feel must be specifically highlighted here.

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

# A. Additional Ablation Studies

In this section, we conduct a comprehensive ablation study of HBCT across varying hyperparameters. Unless stated otherwise, all subsequent experiments use the extended-class settings shown in Table 1.

**Curvature.** In the main paper, we fix the curvature value $K = 1$ following existing literature (Bdeir et al., 2023). We conducted an ablation study to examine how different curvature values affect compatibility performance. Following (Desai et al., 2023), we also evaluate the compatibility performance when we set the curvature value as a learnable parameter, which is adjusted during training with other parameters. The result in Table 3 shows that $K \in [0.5, 1]$ yields stable outcomes in both compatibility and the quality of the new model. With the learnable curvature, the new model showed good performance for new-to-new retrieval. However, it negatively impacted compatibility performance, particularly in cross (new-to-old) retrieval.

| Curvature $K$ | self | cross | $\mathcal{P}_{\text{up}}$ | $\mathcal{P}_{\text{comp}}$ |
|---|---|---|---|---|
| 0.1 | 0.581 | 0.366 | -0.114 | 0.108 |
| 0.5 | 0.651 | 0.397 | -0.008 | 0.206 |
| 0.7 | 0.657 | 0.399 | 0.002 | 0.211 |
| 1.0 | 0.654 | 0.398 | -0.003 | 0.207 |
| 1.5 | 0.604 | 0.370 | -0.079 | 0.123 |
| learnable | 0.660 | 0.371 | 0.006 | 0.124 |

*Table 3.* Ablation of curvature $K$ on HBCT's mAP performance.

**Clipping threshold $\zeta$.** In the main paper, we set the clipping threshold to $\zeta_0 = 1.0$. For the updated model, we raise it to $\zeta_n = \zeta_0 + 0.2$ to accommodate the evolving embedding space and allow new embeddings greater flexibility to diverge from the old ones. This ablation examines the impact of $\zeta_n$: we hold $\zeta_0 = 1.0$ and vary $\zeta_n \in [1.0, 1.3]$. As shown in Table 4, keeping $\zeta_n = 1.0$ preserves the new model's self-performance but harms cross-model compatibility, whereas setting $\zeta_n = 1.2$ offers the best balance between the two metrics.

| $\zeta_o$ | $\zeta_n$ | self | cross |
|---|---|---|---|
| 1.0 | 1.0 | 0.7232 | 0.5525 |
| 1.0 | 1.1 | 0.7153 | 0.5719 |
| 1.0 | 1.2 | 0.7221 | 0.5705 |
| 1.0 | 1.3 | 0.7154 | 0.5588 |

*Table 4.* Effect of the clipping threshold $\zeta_n$ on the CMC@1 performance of the new model.

**Alignment weights.** We evaluate HBCT under a range of hyperparameter settings. Figure 7 shows that setting $\lambda = 0.3$ and $\tau = 0.5$ achieves the best balance between the updated model's self-performance and cross-model compatibility. To assess the relative importance of entailment versus contrastive objectives, we introduce a new weight $\lambda_{\text{entail}}$ into our loss function. In the experiment shown in Figure 7c, we keep $\lambda = 0.3$ fixed and vary $\lambda_{\text{entail}}$.

$$\mathcal{L} = \mathcal{L}_{\text{base}} + \lambda(\mathcal{L}_{\text{entail}} + \mathcal{L}_{\text{contrast}}), \tag{16}$$

**Distance function.** We compare HBCT under three distance functions—geodesic distance (Eq. (3)), squared Lorentz distance (Law et al., 2019), and the Lorentz inner product (Eq. (1))—and we also replace the contrastive loss $\mathcal{L}_{\text{contrast}}$ with a mean-distortion loss

$$\mathcal{L}_{\text{mdist}} = d(\mathbf{h}_o, \mathbf{h}_n),$$

where $d(\cdot, \cdot)$ is either the geodesic or squared Lorentz distance. As Table 5 shows, all three metrics yield similar HBCT performance. Substituting in $\mathcal{L}_{\text{mdist}}$ neither enhances cross-model compatibility nor preserves the updated model's self-performance, echoing the degradation observed with the $\ell_2$ alignment loss in Euclidean space.

**Classification performance.** Throughout the paper, we mainly focus on CMC@k and mAP to focus on the retrieval task. We report the classification accuracy of the alignment methods in comparison with the CMC@1 metric in Table 6.

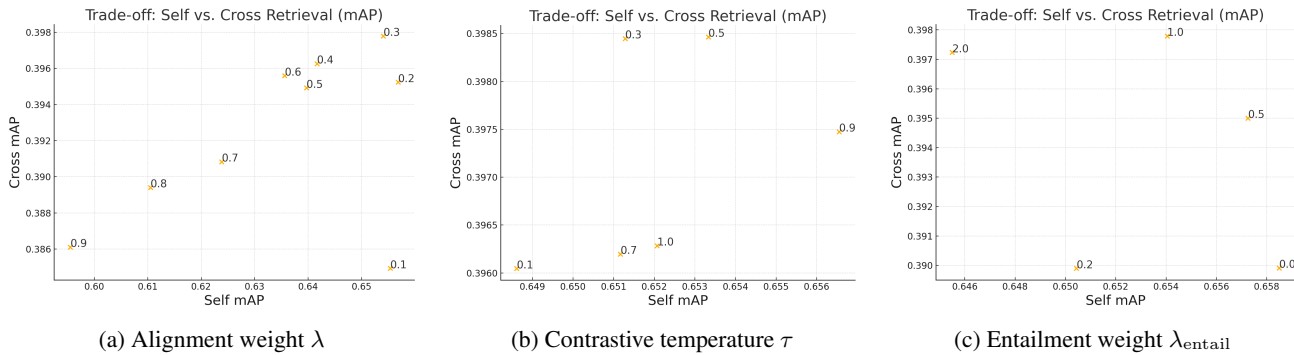

(a) Alignment weight $\lambda$  (b) Contrastive temperature $\tau$  (c) Entailment weight $\lambda_{\text{entail}}$

*Figure 7.* Comparison of self vs. cross retrieval mAP across three hyperparameters.

| Distance function | self | cross |
|---|---|---|
| RINCE - Geodesic | 0.6552 | 0.3982 |
| RINCE - Lorentz inner | 0.6550 | 0.3985 |
| RINCE - Squared Lorentz | 0.6529 | 0.3979 |
| Mean distortion - Geodesic | 0.6487 | 0.3970 |
| Mean distortion - Squared | 0.6381 | 0.3923 |

*Table 5.* Ablation of different distance functions on self and cross mAP performance.

Specifically, we report the cross-classification metric, where the accuracy is evaluated for the old classifier head applied to the new embedding model.

| | Self-CMC@1 | Cross-CMC@1 | Self-Acc | Cross-Acc |
|---|---|---|---|---|
| BCT | 0.695 | 0.447 | 76.47 | 40.66 |
| Hot-refresh | 0.715 | 0.498 | 77.65 | 41.20 |
| HOC | 0.713 | 0.490 | 77.67 | 41.36 |
| HBCT | 0.722 | 0.572 | 78.86 | 42.19 |

*Table 6.* Comparison of different methods on CMC@1 and accuracy metrics.

