# OpenReview forum: "Learning Along the Arrow of Time: Hyperbolic Geometry for Backward-Compatible Representation Learning"
_ICML.cc/2025/Conference — ICML 2025 poster_

### Official Review · Reviewer_JxAG · 2025-02-25

**Overall Recommendation:** 3

**Summary:**

The authors propose Hyperbolic Backward-Compatible Training (HBCT), which is essentially an objective for backwards-compatible representation learning in hyperbolic space. HBCT balances the objective of the embedding loss (e.g. cross-entropy on image classification) with a hyperbolic entailment loss that encourages a partial order between old-model and new-model embeddings, dynamically weighted (via RINCE loss) by the old model’s uncertainty for that input. Across a suite of experiments covering a variety of plausible compatibility scenarios (extending data, extending classes, new architecture, and combinations thereof), HCBT performs well.

## Update after rebuttal
I was satisfied with the authors' responses and am maintaining my weak-accept recommendation.

**Claims And Evidence:**

The authors’ claims about their method’s performance on retrieval tasks are solidly backed by experimental results and theoretical justification. Some of their implementation details (e.g. many hyperparameter choices, the specific formula for uncertainty, etc.) are limited. I call these out more specifically in my questions for the authors.

**Essential References Not Discussed:**

To the best of my knowledge, the authors’ references are adequate.

**Experimental Designs Or Analyses:**

On the basis of the description in the paper, all aspects of the experimental design appear sound to me.

**Methods And Evaluation Criteria:**

The authors’ approach to testing their method makes sense to me, though I would like to see classification accuracy included as a metric in their results (see Weaknesses section).

**Other Comments Or Suggestions:**

* L115: The Lorentzian norm is defined but not used; what the authors are really interested in is the norm of the spacelike dimensions
* L288: The reference to “our proprietary training dataset” should be removed from the paper. The paper should only be evaluated on the merits of the actual experiments presented therein.
* Figure 5: the UMAP embeddings would be more informative if we could see which embeddings are paired with which, e.g. by drawing lines or arrows between old and new embeddings. As this will likely create a lot more visual noise, it may necessitate reducing the total number of points visualized.
* I caught a couple of typos:
  * R253: “making it adaptively” should be “adaptive”
  * R270: empty parentheses
  * R371: “can be achieved only minimal” should be “can be achieved with only minimal”
  * L431: “aligns with out intuition” should be “aligns with our intuition”

**Other Strengths And Weaknesses:**

**Strengths:**
* **Novelty:** this is the first work (to my knowledge) to combine the fields of backward-compatible representation learning and hyperbolic geometry; the motivation for combining these is compelling. The use of entailment loss as a relaxation of distance-based matching for embeddings is especially insightful.
* The authors demonstrate strong results on a wide range of retrieval benchmarks. This is an admittedly idealized setting, but it gets at the core differences between their approach and other approaches in the literature.
* The method is highly general: in particular, the models are not expected to be hyperbolic to start with.

**Weaknesses:**
* **Many choices are unjustified:**
  * Most hyperparameter choices are only justified in terms of existing literature. The authors claim to have tried a hyperparameter sweep converging on $\lambda=0.3, \tau=0.5, \beta=0.01$, but do not include the results in the Appendix.
  * The use of the Lorentz model is somewhat surprising, given that the entailment cone literature generally relies on the Poincare model and the authors make no use of the timelike dimension. I understand that these models are interchangeable, so it may be the case that this is purely a matter of convenience/numerical stability—if so, I would like the authors to clarify this.
  * The derivation of Equation 10 is unclear: what is the relationship between this quantity and the distance from the origin of the hyperboloid?
* **Poor structure** throughout the work. For instance:
  * Sections 2.1 and 2.2 could be combined (there is no need to e.g. define the tangent plane twice);
  * It is unclear why "norm control for numerical stability" and "overall training pipeline" are under subsection 4.3;
* **Limited results:** The authors only test retrieval performance, evaluated via CMC@k and mAP. While I agree these are the most important things to test, what other metrics can be evaluated? In particular, since the authors discuss classification as a training objective, what happens to classification performance?

**Questions For Authors:**

* In your paper, you say "their exponential growth of areas and volumes with respect to time makes them particularly suited for applications involving continual model updates with new entities or classes arrive over time." Intuitively, what is the connection between exponential growth in neighborhoods and applications with continual model updates? I could just as easily believe, for instance, that continuous model updates work fine in polynomially-growing vector spaces.
* Can this approach be used to train a new model to align embeddings between several different models? Assume we don’t know anything about how they were trained
* What happens if you ablate the hyperbolic MLR (i.e. compute loss on Euclidean model, then exp map, then compute the other losses)—does this do better or worse?

**Relation To Broader Scientific Literature:**

This paper unites the fields of hyperbolic deep learning (particularly the computer vision applications thereof) with backwards-compatible learning. To the best of my knowledge, this is the first paper at the intersection of these two fields.

**Theoretical Claims:**

The authors provide no proofs with their paper. For most of their theoretical claims, the authors defer to the existing literature, which is fine by me. However, certain claims—the form of equation 10, the use of the hyperboloid model, the choice of $\beta_n = \beta_o + 0.2$ for clipping—are neither justified with proofs/mathematical intuitions nor tested via ablations.

---

> ### Author Rebuttal · Authors · 2025-04-01
>
> We thank the reviewer for the constructive feedback and suggestions. We answer the remaining questions by points as follows.
>
> > Q1. Ablation on hyperparameter sweeping
>
> Due to restricted space, we report the ablation for the clipping threshold. We refer to our response to Reviewer fRer for the ablation with the curvature value. Remaining hyperparameter ablations (alignment weight, entailment cone weight, temperature) are included in the revision.
>
> | $\beta_n = \beta_o + \zeta$ |self-CMC@1|cross-CMC@1|
> |-|-|-|
> | $\zeta = 0.0$| 0.7232 | 0.5525  |
> | $\zeta = 0.1$| 0.7153 | 0.5719 |
> | $\zeta = 0.2$| 0.7221 | 0.5705 |
> | $\zeta = 0.3$| 0.7154 | 0.5588 |
>
> > Q2. The use of the Lorentz model… it may be the case that this is purely a matter of convenience/numerical stability—if so, I would like the authors to clarify this.
>
> Yes, we use the Lorentz model for its numerical stability, following recent work [1, 2]. Its space-time interpretation also fits our context, allowing us to use the time-like dimension to model continual updates and embedding uncertainty.
>
> > Q3. The derivation of Equation 10 is unclear: what is the relationship between this quantity and the distance from the origin of the hyperboloid?
>
> We define Lorentz uncertainty via its isometric relation to the Poincaré model. Both share the same form, based on the Euclidean norm of the pre-exponential embedding.
> $$
> 	\mathrm{Uncertainty}(\mathbf{h}^{\mathcal{L}}) = \mathrm{Uncertainty}(\mathbf{h}^{\mathbb P}) = 1 - \frac{1}{\sqrt{K}} \mathrm{tanh}(\sqrt{K} \|\| \mathbf z \|\|)
> $$
>
> The distance from the origin of the hyperboloid is given by
>
> $$
> 	d(\bar{\mathbf{0}}, \mathbf{h}) = \mathrm{cosh}^{-1} (\frac{1}{\sqrt{K}} cosh(\sqrt{K} \| \| \mathbf z \| \|)
> $$
> Both the distance to the origin and our uncertainty measure depend explicitly on the Euclidean $\ell_2$-norm before the exponential map. While prior work uses the unbounded origin distance as a proxy for uncertainty, our bounded [0,1] measure is more interpretable and manageable.
>
> > Q4. What other metrics can be evaluated? classification performance?
>
> We report CMC@1 and mAP to focus on retrieval performance. The table shows accuracy and retrieval results for various alignment methods, including cross-accuracy using the old classifier on new features. Accuracy trends generally follow retrieval, though with smaller gaps.
>
> || Self-CMC@1 | Cross-CMC@1 | Self-Acc | Cross-Acc|
> |-|-|--|-|-|
> |BCT | 0.695 | 0.447  | 76.47  | 40.66|
> |Hot-refresh | 0.715 | 0.498 | 77.65  | 41.2|
> |HOC | 0.713 | 0.490 | 77.67 | 41.36|
> |HBCT | 0.722 | 0.572 | 78.86  | 42.19|
>
> > Q5. How does exponential volume growth relate to continual model updates, and why might it be preferable to polynomially growing spaces like Euclidean space?
>
> We argue that exponential volume growth in hyperbolic space allows more room to represent new entities with smaller increases in radius, which is advantageous for continual updates. In contrast, Euclidean space requires rapidly growing radii to accommodate new entities.
>
> Moreover, Lemma 1 in [3] shows that in Euclidean space, the probability of new class prototypes aligning with a trained model decreases exponentially with the number of dimensions and classes, leading to an impossibility result for backward compatibility. We hypothesize that hyperbolic geometry with its exponential growth property may mitigate this issue, though a formal analysis is left for future work.
> > Q6. Can this approach be used to train a new model to align embeddings between several different models? Assume we don’t know anything about how they were trained.
>
> Aligning a new model to multiple old models is challenging without knowing their training processes. Some models may have incompatible embeddings, enforcing inherent trade-offs in the compatibility. A common solution is to add a projection layer to map embeddings into a shared space [4], which is orthogonal and can be integrated to our method.
>
> > Q7. What happens if you ablate the hyperbolic MLR (i.e. compute loss on Euclidean model, then exp map, then compute the other losses)—does this do better or worse?
>
> We replaced the hyperbolic MLR with its Euclidean counterpart while keeping the entailment cone and RINCE loss in hyperbolic space. This led to a significant drop in both the new model's performance and its compatibility. This is likely due to conflicting signals between the Euclidean MLR and hyperbolic RINCE loss.
>
> |  | Self-mAP | Cross-mAP |
> |-|-|-|
> |HBCT w/o entail  | 0.435    | 0.388|
> |HBCT  | 0.655 | 0.398|
> |HBCT-EMLR w/o entail | 0.4791   | 0.047|
> |HBCT-EMLR   | 0.4897  | 0.279|
>
> > Q8. Writing structure and  typos
>
> We reorganized the "norm control" and "overall training" into a separate section and fixed the typos.
>
> References:
>
> [1] Bdeir et al. Fully Hyperbolic Convolutional…, ICLR’24.
> [2] Desai et al. Hyperbolic Image-Text…, ICML’23.
> [3] Biondi et al. Stationary Representations: Optimally…, CVPR’24.
> [4] Linia et al. Asymmetric Image Retrieval with Cross Model...

---

> > ### Comment · Reviewer_JxAG · 2025-04-01
> >
> > I thank the authors for addressing my questions thoroughly. Overall, I found their additional experiments and clarifications helpful for evaluating the paper.
> >
> > I intend to keep my already positive rating of 3. I believe this work introduces a valuable approach combining hyperbolic geometry with backward compatibility. I encourage the authors to add the classification metrics to the final version of this paper, as these strengthen the case for their model.

---

> > > ### Author Response · Authors · 2025-04-02
> > >
> > > We sincerely thank the reviewer for their thoughtful feedback and positive assessment of our work. We are glad that our additional experiments and clarifications were helpful in evaluating the paper. We will make sure to incorporate additional experiments and discussion into the final version of the paper.
> > >
> > > Best,
> > > Authors

---

### Official Review · Reviewer_eHD4 · 2025-03-14

**Overall Recommendation:** 3

**Summary:**

This paper aims to improve backward compatible representation learning by using hyperbolic embeddings instead of Euclidean embeddings. This paper claims that using hyperbolic embeddings achieves greater compatibility with previous models without compromising the performance of new embedding models.

Methods-wise, this paper uses a hybrid Euclidean-Hyperbolic model to encode images in hyperbolic space. This allows better integration of existing Euclidean encoders compared to fully hyperbolic models. The paper also defines an uncertainty measure for Lorentz spaces, based on the analogous uncertainty measure for Poincare spaces. The model uses two auxiliary losses, an entailment cone loss, and RINCE-based loss, which is modified to include the uncertainty measure defined in the paper.

**Claims And Evidence:**

The high-level intuition of the method is interesting and makes a lot of sense.

The authors claim that HBCT enhances backward compatibility without compromising the performance of new embedding models. This is supported by the empirical evidence, which suggests that HBCT enhances backward compatibility (best $P_\mathrm{com}$ scores in Table 1) with little tradeoff in performance of the new models ($P_\mathrm{up}$ scores near 0 in Table 1).

The use of the entailment loss and the RINCE-based loss both seem effective according to the ablations (Table 2), but the performance of RINCE is a bit more mixed, sometimes resulting in a large drop in compatibility.

**Essential References Not Discussed:**

N/A

**Experimental Designs Or Analyses:**

Overall the experimental design seems reasonable. However, this paper could benefit from analyses on more datasets than just CIFAR-100 and TinyImageNet.

**Methods And Evaluation Criteria:**

Overall, the methods and evaluation criteria seem reasonable.

It is not clear to me why the Lorentz uncertainty was defined the way it was. Perhaps the connection to Poincare uncertainty could be made more explicit in the text.

Although not the main focus of the paper, I think it is still important to also report the performance of all models on CMC@1 and mAP directly so readers can assess the tradeoff between compatibility and performance.

**Other Comments Or Suggestions:**

Equation 12, the apostrophe in $h_o’$ not defined
Missing citation (Line 269, second column)

**Other Strengths And Weaknesses:**

Weaknesses

In Table 1, the self and cross columns are not explained or referred to in the main text.

It seems that a possible limitation of the model is that performance may degrade after many model updates. It would be good if the authors could discuss this.

**Questions For Authors:**

1. How was the Lorentz uncertainty derived?
2. What is the CMC@1 and mAP performance of the models?
3. What do the self and cross columns mean in Table 1?
4. Is there a reason why using the RINCE-based loss sometimes results in a large drop in compatilibity?
5. Given how the norms grow with repeated model updates and the possible resulting instability, how does the performance of this method compare to Euclidean methods after many model updates?

**Relation To Broader Scientific Literature:**

The idea of using hyperbolic embeddings for backwards compatibility is novel and is also a very natural idea.

The method of this paper is a combination of existing methods. The hybrid Euclidean-Hyperbolic model is based on [1], and the entailment cone loss was first proposed in [2]. The RINCE-based loss is essentially the RINCE loss of [3] with the modification that the $q$ parameter is the Lorentz uncertainty measure. In light of this, the novelty of the paper's methods is quite limited.

[1] Khrulkov, Valentin, et al. "Hyperbolic image embeddings." Proceedings of the IEEE/CVF conference on computer vision and pattern recognition. 2020.

[2] Desai, Karan, et al. "Hyperbolic image-text representations." International Conference on Machine Learning. PMLR, 2023.

[3] Chuang, Ching-Yao, et al. "Robust contrastive learning against noisy views." Proceedings of the IEEE/CVF conference on computer vision and pattern recognition. 2022.

**Theoretical Claims:**

N/A

---

> ### Author Rebuttal · Authors · 2025-04-01
>
> We thank the reviewer for the constructive feedback and suggestions. We answer the remaining questions by points as follows.
>
> > Q1: why the Lorentz uncertainty was defined the way it was.
>
> We define the Lorentz uncertainty using the isometric relation between the Poincaré and Lorentz models. Both uncertainty measures in Poincaré and Lorentz models have the same form with respect to the norm 2 of the Euclidean embedding before the exponential map:
>
> $$
> 	\mathrm{Uncertainty}(\mathbf{h}^{\mathcal{L}}) = \mathrm{Uncertainty}(\mathbf{h}^{\mathbb P}) = 1 - \frac{1}{\sqrt{K}} \mathrm{tanh}(\sqrt{K} \|\| \mathbf z \|\|)
> $$
> Both the distance to the origin and our uncertainty measure depend explicitly on the Euclidean $\ell_2$-norm before the exponential map. While prior work uses the unbounded origin distance as a proxy for uncertainty, our bounded [0,1] measure is more interpretable and manageable. We will discuss this explicitly in the revision.
>
> > Q2+3: What is the CMC@1 and mAP performance of the models?… What do the self and cross columns mean in Table 1?
>
> The *self* (new-to-new retrieval) and *cross* (new-to-old retrieval) columns in the Table refer to the original model performance (CMC or mAP). We thank the reviewer for pointing this out, and we will explain this explicitly in the paper to avoid future confusion.
>
> > Q4: Is there a reason why using the RINCE-based loss sometimes results in a large drop in compatibility?
>
> During the experiments, we observed that training with RINCE-based loss sometimes is more difficult to converge in some cases, especially on the Vision Transformer model. This maybe because of the instability of the exponential map when applied to the large norm produced by ViT. However, when training with the entailment cone loss, the training becomes more stable.
>
> > Q5: Given how the norms grow with repeated model updates and the possible resulting instability, how does the performance of this method compare to Euclidean methods after many model updates?
>
> We conducted an experiment with five consecutive model updates to test the performance of different compatibility methods on the CIFAR100. The training data will be split into five sets, each containing 20, 40, 60, 80, and 100 (all) classes, respectively. For the first two models (time steps 0, 1), we will use ResNet18 as the base model and we will update to ResNet50 at the last three steps (2, 3, 4). For each method, we report the CMC@1 compatibility matrix for HBCT, HOC and BCT. Columns are the encoders for the query, and rows are the encoders for the gallery set. It can be seen that the HBCT can maintain the compatibility with the old model after several updates ($\phi_1 / \phi_4$) = 0.32) , while HOC and BCT quickly deteriorate after a few model updates ($\phi_1 / \phi_4$ = 0.207 for HOC and 0.14 for BCT).
>
> HBCT
>
> |  $\phi_o / \phi_n$    | $\phi_0$ | $\phi_1$ | $\phi_2$ | $\phi_3$ | $\phi_4$ |
> |-|-|-|-|-|-|
> | $\phi_0$ | 0.2016   | | | | |
> | $\phi_1$ | 0.2295   | 0.3585   ||||
> | $\phi_2$ | 0.2585   | 0.4309   | 0.4872   |||
> | $\phi_3$ | 0.2867   | 0.4857   | 0.5525   | 0.6166   | |
> | $\phi_4$ | 0.3261   | 0.5227   | 0.6051   | 0.6995   | 0.7424   |
>
> HOC
> |    $\phi_o / \phi_n$        | $\phi_0$ | $\phi_1$ | $\phi_2$ | $\phi_3$ | $\phi_4$ |
> |-|-|-|-|-|-|
> | $\phi_0$ | 0.1825   | | | | |
> | $\phi_1$ | 0.17     | 0.3265   | | | |
> | $\phi_2$ | 0.1954   | 0.3807   | 0.4831   | | |
> | $\phi_3$ | 0.207    | 0.418    | 0.5357   | 0.6045   | |
> | $\phi_4$ | 0.1926   | 0.4332   | 0.5831   | 0.6888   | 0.7302   |
>
> BCT
>
> |  $\phi_o / \phi_n$      | $\phi_0$ | $\phi_1$ | $\phi_2$ | $\phi_3$ | $\phi_4$ |
> |-|-|-|-|-|-|
> | $\phi_0$ | 0.1818   |          |          |          |          |
> | $\phi_1$ | 0.1589   | 0.3124   |          |          |          |
> | $\phi_2$ | 0.1524   | 0.3218   | 0.4456   |          |          |
> | $\phi_3$ | 0.135    | 0.3299   | 0.4834   | 0.5647   |          |
> | $\phi_4$ | 0.1497   | 0.3212   | 0.5249   | 0.5939   | 0.7206   |
>
> > Q6: It seems that a possible limitation of the model is that performance may degrade after many model updates.
>
> Yes, we have discussed this in the limitations section. Increasing the clipping threshold after each update can lead to numerical instability when it becomes large (e.g., >10). One potential solution is rescaling the time dimension of all embeddings after a certain number of updates (e.g., >20). Another approach is online backfilling, where a subset of items is used to update the gallery set—this can help manage the growing time dimension.
>
> We hope that we have addressed all your concerns adequately. We have updated the manuscript according to your comments. Please let us know if we can provide any further details and/or clarifications.

---

### Official Review · Reviewer_fRer · 2025-03-18

**Overall Recommendation:** 4

**Summary:**

The paper proposes to leverage hyperbolic geometry for backward compatible learning: setting in which a model is updated and its representation should preserve compatibility with representations from the model before the update. The authors propose a  loss composed of  two terms to (i) constraint the new embedding to lie in the entailment cone of the old embeddings (ii) regularization loss which align weights according to the uncertainty, to preserve performance of the new model.The method is validated using the CIFAR100 and MiniImagenet datasets, analyzing backward compatibility when introducing novel data samples, novel classes or a different architecture. Ablation experiments are performed to quantify the contribution of each term and setting. The method outperforms existing methods which are based on considering euclidean geometry in representation space.

**Claims And Evidence:**

To the best of my judgment,  claims in the paper are validate in the paper. In particular:

(i) They show how hyperbolic geometry is good modeling choice for the problem of backward compatibility, showing previous contains as the entailment loss and the uncertainty estimation naturally fits into this problem. The is further confirmed by the experiments in Table 2.

(ii) The experiments demonstrate how hyperbolic geometry is a better fit than exiting methods based on euclidean geometry.

**Essential References Not Discussed:**

To the best of my knowledge there are no very fundamental related work that has not been cited. It could be  beneficial to include the references in the previous section, although this is not strictly necessary.

**Experimental Designs Or Analyses:**

## Strenghts

 - Experiments are sounds and shows that under different setting (sample change, class change, and architectural change) that hyperbolic spaces are effective for the problem of backward compatibility.

- Ablations show the impact of each term in the loss, providing a good understanding of the method.


## Weaknesses

- Despite the good performance demonstrated on the proposed benchmark, previous works introducing the problem of backward compatibility (Shen et al 2020) experimented with larger models (e.g. Resnet 100) and in particular larger datasets (e.g. IMDB). This raise some questions on how the performance of the proposed method would scale to this size.


- Some details in the table of result could be explained more in depth (e.g. cross and self columns, see questions section)

**Methods And Evaluation Criteria:**

The paper compares to many baselines, according to the best of my knowledge.  Concerning the dataset, in the paper (Shen et al 2022) introducing backward compatibility larger models and datasets (e.g. Imdb ) were adopted, so it would be interesting to see if the result scales up to larger datasets.

**Other Comments Or Suggestions:**

I spotted the following typos:


Line 270: citation missing: "unlike previous methods (),"

Line 312: "the the" -> "the"

**Other Strengths And Weaknesses:**

### Strenghts

- *Clarity* the paper is very clear and well written.

- *Originality* the work result in the effective combination existing frameworks for robust contrastive loss (Chuang et al., 2022) , uncertainty properties of hyperbolic geometrical representation spaces (Franco et al., 2023; Atigh et al., 2022) applied to backward compatibility (Shen et al 2022), resulting in an effective and good fitting framework for the problem.

- *Significance*: the problem of backward compatibility I still very open and recent, and of importance for practical application (.e.g industrial deployment of models). The paper demonstrate the importance of correctly characterizing the geometry of representational space, with important practical consequences. Moreover the type of research has possible implication on  diverse fields such as representational alignment, model adaptation, test time training, and out of distribution detection.


## Weaknesses

- Limitations in assuming hyperbolic geometry: the hyperbolic geometry assumption could be also limiting in my understanding in two ways: training could be more expensive and applicability of the method could be limited as most methods are not assumed to be trained with hyperbolic embeddings spaces. Also some questions arise when the geometry of the space is not flat and nor with constant curvature.  On the regard see last two questions in the question section.

- A discussion and comparison with methods that use product of spheres seems to miss.

**Questions For Authors:**

- How does the similarity measure is used to compare queries to gallery samples in the Cumulative Matching Characteristics metric for retrieval affect results? What happens if one use a different measure (eg. euclidean distance in euclidean based method, or hyperbolic geodesic distance in euclidean based methods, assuming despite pertaining an hyperbolic geometry)

- In Table 1 the columns corresponding to self and cross denote the base loss? what is this number referring to, the original model performance?

- The authors mention approaches that use hypersphere geometry (with cosine similarity). What's the comparison with these methods?

- How does curvature K affect the performance? why the choice of fixing it to 1 for all experiments?

-  How much impactful is pretraining from scratch assuming hyperbolic geometry? How does this compare to finetune from a pretrained model assuming hyperbolic geometry as opposed to train from scratch? Although this is not the direct focus of the paper, the applicability of the method for backward compatible training to pretrained models is an important point in order to apply the method.

**Relation To Broader Scientific Literature:**

Key contribution of there paper are to demonstrate the importance of characterizing the geometry of representational space with a different metric than the euclidean, i.e. flat. Previous work has highlighted the importance of characterizing different metrics in latent spaces in distinct setting, e.g. generative [2], text [3,4], images (Khrulkov et al., 2020). To the best of my knowledge this is  the first work that tries to advocate for a different geometry to solve the recent problem of backward compatibility, demonstrating an elegant and effective solution.


The work has also important consequences and relation to the field of representation alignment [e.g. 1,5] which could be interesting to relate to.

_[1] Moschella, Luca, et al. "Relative representations enable zero-shot latent space communication." ICLR 2023_

_[2] Arvanitidis, Georgios, Lars Kai Hansen, and Søren Hauberg. "Latent space oddity: on the curvature of deep generative models._

_[3] Gu, Albert, et al. "Learning mixed-curvature representations in product spaces." International conference on learning representations. 2018._

_[4] Dhingra, Bhuwan, et al. "Embedding text in hyperbolic spaces., ACL_

_[5] Huh, Minyoung, et al. "Position: The platonic representation hypothesis." Forty-first International Conference on Machine Learning. 2024._

**Theoretical Claims:**

No theoretical claims are present.

---

> ### Author Rebuttal · Authors · 2025-04-01
>
> We thank the reviewer for the constructive feedback and suggestions. We answer the remaining questions by points as follows.
>
> > Q1: Training could be more expensive with hyperbolic geometry
>
> We agree hyperbolic models may be more expensive, but our method only adds a simple exponential map to a Euclidean model, incurring minimal overhead. We report the running time of ResNet18 with a batch size of 256 in the table below.
>
> | ResNet18  | Forward only | Forward & Backward |
> |-|-|-|
> | Euclidean | 7.9e-07|  0.0556|
> | Euclidean-Hyperbolic | 8.8e-07 | 0.0630|
>
> > Q2: Applicability could be limited as most methods are not assumed to be trained with hyperbolic embeddings spaces.
>
> Our method is general and can be applied on top of existing Euclidean embedding models.
>
> > Q2: Some questions arise when the geometry of the space is not flat and nor with constant curvature
>
> We consider Euclidean and hyperbolic geometries because they are most commonly used in existing retrieval systems, mostly because the geodesic distance between two points can be computed efficiently. In some cases, manifolds with non-constant curvature can better represent data, but the geodesic distance will be more computationally expensive, making it difficult to generalize to large-scale retrieval systems.
>
> > Q3: The comparison with these approaches that use hyperspherical geometry (with cosine similarity)?
>
> Hot-Refresh and HOC are two alignment methods that employ hyperspherical geometry, as they align models using a contrastive objective based on cosine similarity. Since cosine similarity normalizes embeddings to have unit norm ($| z |_2 = 1$), these methods map embeddings onto a shared unit hypersphere. Pairwise embedding comparisons are then computed using the negative inner product as a distance measure. We will clarify this in the baseline discussion in the revision.
>
> > Q4: How does the similarity measure used in CMC affect results?
>
> We observe that retrieval performance remains consistent across different similarity measures. Below are the results on CIFAR100:
> - Euclidean-ResNet18:
>   - Cosine: mAP 0.6543, CMC@1 0.7157
>   - Euclidean: mAP 0.6543, CMC@1 0.7157
> - Hyperbolic-ResNet18:
>   - Geodesic: mAP 0.6548, CMC@1 0.7177
>   - Lorentz inner product: mAP 0.6548, CMC@1 0.7177
> - Using hyperbolic distance on Euclidean embeddings (via exponential map):
>   - Hot-refresh original:
>      - new-to-new retrieval (self): mAP: 0.649485, CMC@1: 0.7147
>      - new-to-old retrieval (cross): mAP: 0.347584, CMC@1: 0.4977
>   - Hot-refresh -> exponential map
>      - new-to-new retrieval (self): mAP: 0.649482, CMC@1: 0.7147
>      - new-to-old retrieval (cross): mAP: 0.347583, CMC@1: 0.4978
>
> > Q5: In Table 1 the columns corresponding to self and cross denote the base loss? what is this number referring to, the original model performance?
>
> Yes, *self* (new-to-new retrieval) and *cross* (new-to-old retrieval) refer to the original model performance (CMC or mAP). We will explain this explicitly in the paper to avoid future confusion.
>
> > Q6. Experiments larger datasets
>
> We mainly followed [1, 2] to test on CIFAR100 and Tiny-ImageNet, where hyperbolic geometry is shown to be effective. Given the short time frame for rebuttal responses, extensive experiments on large datasets are beyond our immediate scope but we acknowledge the valuable suggestion and plan to explore larger-scale evaluations in future research to further validate the approach.
>
> > Q7: How does curvature K affect the performance? why the choice of fixing it to 1 for all experiments?
>
> We set a fixed curvature value following the existing literature [1]. We conducted an ablation study to examine how different curvature values affect compatibility performance. Generally,  $K \in [0.5, 1.0]$ yields stable outcomes in both compatibility and the quality of the new model. When we also tested the learnable curvature, it showed good performance in the new model but negatively impacted compatibility performance.
>
> |Curvature K|self (mAP)|cross (mAP)|P_up (mAP)|P_comp (mAP)|
> |-|-|-|-|-|
> |0.1 |0.581|0.366|-0.114|0.108 |
> |0.5 |0.651|0.397|-0.008 |0.206 |
> |0.7|0.657 |0.399 |0.002|0.211 |
> |1 |0.654|0.398 |-0.003|0.207|
> |1.5|0.604 |0.370|-0.079 |0.123|
> |Learnable  |0.660 |0.371|0.006|0.124|
>
> > Q8: How impactful is pretraining from scratch assuming hyperbolic geometry? How does this compare to finetuning from a pretrained model assuming hyperbolic geometry as opposed to training from scratch?
>
> In our experiment, ResNet18 is trained from scratch, while ViT is finetuned from ImageNet21K pretrained weights to reflect a typical usage of open-source models. We do not pretrain ViT from scratch due to the poor convergence on CIFAR100 and TinyImagenet.
>
> References:
> [1] Bdeir et al. Fully Hyperbolic Convolutional Neural Networks for Computer Vision, ICLR’24.
> [2] Biondi et al. Stationary Representations: Optimally Approximating Compatibility and Implications for Improved Model Replacements, CVPR'24.

---

### Decision · Program_Chairs · 2025-05-01

**Decision:**

Accept (poster)

**Comment:**

This paper investigates backward-compatible representation learning, an important aspect of real-world deployment of neural networks. The paper advocates for hyperbolic geometry to address this problem, to deal with uncertainty and model quality. After the rebuttal and discussion phases, all three reviewers vote towards accept. The reviewers find the motivation interesting and the paper novel. They also point out fair concerns regarding the experimental scope of the paper. The rebuttal does a good job in providing additional metrics and ablations, but the AC agrees with the reviewers that the model and dataset sizes remain small-scale. Regardless, the AC find the paper novel and clear. The AC agrees with the unanimous consensus for accept.